# Mutant-Huntingtin Molecular Pathways Elucidate New Targets for Drug Repurposing

**DOI:** 10.3390/ijms242316798

**Published:** 2023-11-27

**Authors:** Vladlena S. Makeeva, Nadezhda S. Dyrkheeva, Olga I. Lavrik, Suren M. Zakian, Anastasia A. Malakhova

**Affiliations:** 1Institute of Cytology and Genetics, Siberian Branch of Russian Academy of Sciences, 10 Akad. Lavrentiev Ave., 630090 Novosibirsk, Russia; vladamakkeeva@gmail.com (V.S.M.); zakian@bionet.nsc.ru (S.M.Z.); amal@bionet.nsc.ru (A.A.M.); 2Institute of Chemical Biology and Fundamental Medicine, Siberian Branch of Russian Academy of Sciences, 8 Akad. Lavrentiev Ave., 630090 Novosibirsk, Russia; lavrik@niboch.nsc.ru

**Keywords:** Huntington’s disease, drug target, PARP1 inhibitor, drug repurposing

## Abstract

The spectrum of neurodegenerative diseases known today is quite extensive. The complexities of their research and treatment lie not only in their diversity. Even many years of struggle and narrowly focused research on common pathologies such as Alzheimer’s, Parkinson’s, and other brain diseases have not brought cures for these illnesses. What can be said about orphan diseases? In particular, Huntington’s disease (HD), despite affecting a smaller part of the human population, still attracts many researchers. This disorder is known to result from a mutation in the *HTT* gene, but having this information still does not simplify the task of drug development and studying the mechanisms of disease progression. Nonetheless, the data accumulated over the years and their analysis provide a good basis for further research. Here, we review studies devoted to understanding the mechanisms of HD. We analyze genes and molecular pathways involved in HD pathogenesis to describe the action of repurposed drugs and try to find new therapeutic targets.

## 1. Introduction

Various approaches to drug research and development are basically de novo drug discovery and drug repurposing. While the former requires long-term research, the latter is being actively implemented for the treatment of many diseases, owing to the accumulation of ample data in databases and the discovery of mechanisms that unite various diseases. In fact, the accumulated knowledge contributes to the search for therapies for diseases with still poorly understood mechanisms. Thus, drug repurposing is actively used for the treatment of different types of cancer and common neurodegenerative disorders such as Alzheimer’s and Parkinson’s disease [1]. This strategy involves extrapolation of already discovered molecular processes and drug targets to hallmarks and symptoms of a disease in question. In recent studies, for example, PARP inhibitors were often used as repositioned drugs. They had been developed as anticancer drugs, but now researchers test them on other diseases, including Parkinson’s, Alzheimer’s, Huntington’s, and other brain pathologies [2,3]. Huntington’s disease (HD), despite its extrinsic simplicity—only mutation in one gene underlies the pathogenesis—has no effective treatment today. Designing an anti-HD therapy that would eliminate etiology is a topical and relevant task of biomedicine.

In this work, we review some aspects of the latest approaches to drug discovery with an emphasis on repurposing. One of the key objectives is the drug target search. We distinguish several essential ways to search. The first one is driven by hallmarks of the disease, such as protein aggregates or oxidative stress, so that these problems can be resolved by similar medication even if they have different root causes. The second way is to employ certain biomolecules as targets. Nonetheless, in different diseases, the same biomolecule may play different roles in pathogenesis, and hence the effect of the drug may be unexpected. The third way is to target a molecular pathway, not a specific molecule. Within the framework of this field, we discuss issues of drug repurposing and methods of target searches and outline a combined strategy. Another objective of our work is to look for genes of proteins that interact with mutant huntingtin (mHTT) and to determine their molecular pathways. Next, we use the found genes as targets for drug discovery among existing therapeutics. These drugs may be considered subjects of further experiments and potential therapies for HD.

## 2. HD Mechanisms

The development of HD is associated with the expansion (multiplication) of trinucleotide CAG in the first exon of the huntingtin (*HTT*) gene [4]. The pathogenesis of the disease is based on the expression of the huntingtin protein having an abnormal conformation and of its derivatives, such as clusters of N-terminal fragments and protein aggregates [5,6]. The formation of protein bodies driven by various forms of the mutant huntingtin occurs in medium spiny neurons during the progression of the pathology and leads to their death. These phenotypic markers can be observed in postmortem brain samples from patients or animal models [7]. The disease is accompanied by behavioral symptoms, for instance, impaired motor functions and disturbances of mental and cognitive abilities [8].

What happens in HD at the molecular level? At the moment, aberrations are being revealed in such molecular pathways as the utilization of proteins and cellular structures (namely, the ubiquitin–proteasome system and autophagy), expression regulation of intracellular signaling, cytoskeleton-associated processes (e.g., cell transport and division), glutamatergic and dopaminergic synaptic signaling systems, and, finally, mitochondrial function; the latter dysfunction is combined with destabilization of energy homeostasis and oxidative stress, which leads to activation of pathways of a stress response and apoptosis [9,10].

### 2.1. Transcription Dysfunction

The most frequently, as shown by recent studies, transcriptional impairment affects neurotransmitter receptors, ion channels, and BDNF [11,12]. Aberrant huntingtin interacts with different transcription-regulating proteins (p53, CREB, CBP, and MSK-1, which control cell proliferation and DNA damage repair), PGC-1α, organelle and vesicle transporters, and dopamine 2 receptor–interacting proteins [13]. There is evidence that mHTT’s interactions with the transcription factors p53, CBP, CREB, SP1, NF-κB, REST, Foxp1, and HSF1 impede gene expression [14,15,16,17,18,19].

Normally, huntingtin binds to transcription factors that recognize a neuron-restrictive silencer element (NRSE) region of the genes necessary for life support of spiny neurons and thereby regulates their transport to the nucleus from the cytoplasm and their activity. Impairment of this interaction leads, in particular, to the shutdown of expression of the NRSE-containing genes *BDNF* and *REST* [20].

PGC-1α is involved in metabolic regulation and mitochondrial biogenesis. CREB-mediated transcription of PGC-1α is suppressed by mutant huntingtin. Conversely, recovery of the PGC-1α level impedes mHTT aggregation in striatal neurons [21,22]. mHTT inhibits SP1-dependent transcription as shown in postmortem brain samples from HD patients [23]. Additionally, mHTT mediates translocation of the REST protein to the nucleus and causes silencing of genes [24]. The interaction of mHTT with p53 leads to the expression of apoptotic genes [25].

Decreased activity of enzymes that promote chromatin remodeling has been demonstrated in HD. Phosphorylated CREB (a cAMP-dependent transcription factor) binds to a CRE region in the promoter of genes important for neuronal survival [26] and activates transcriptional coactivator CBP, thereby directing it to chromatin being remodeled to assemble a transcription factor complex. mHTT interacts with CBP and TAFII130, thus preventing their binding to the CRE region and interrupting the CRE-mediated transcription process [20,27].

In a recent paper, a set of genes differentially expressed in HD was presented [28]. As for human studies, investigators have found IGDCC3, a participant in nervous system development, and XKR4, a regulator of the apoptotic process during development. In postmortem brain samples, DNAJB1, HSPA1B, and HSPB1 are reported to be upregulated in all cell types except for medium spiny neurons. At the earliest stages of HD, there is a change in the expression of genes *HSPH1* and *SAT1*. Differential expression analysis of genes of transcription regulators has revealed the upregulated factors TCF4, FOSL2, BCL6, ZBTB16, FOXP1, KLF15, RXRA, CUX1, CREBBP, and NFIA and the downregulated factors ZKSCAN1, MAX, E2F3, BCL11A, EGR1, FOXG1, TP53, HMG20A, JMJD1C, and STAT4 [28].

### 2.2. Systems of Clearance of Proteins and Other Cell Components

Heat shock proteins such as HSP70, HSP40, and HSP90 participate in the pathological processes of HD. It has been established in mouse models of HD that an increase in Hsp70 expression plays a neuroprotective role, and inhibition of Hsp90 results in mHTT degradation [29]. HSP40 and HSP70 suppress the formation of fibrillar aggregates from mutant forms of huntingtin, thus facilitating their refolding or, more often, their being directed to the ubiquitination and degradation system of the 26S proteasome complex [29,30,31,32]. Nevertheless, at the same time, heat shock proteins often lose their ability to perform their functions and get mixed up in growing mHTT aggregates. In this case, hydrolysis of the mutant protein by proteases occurs, which leads to the formation of N-terminal toxic fragments of mHTT [33,34]. These fragments exist in several forms: soluble monomers, oligomers, or aggregates. Therefore, mHTT generally disrupts the process of protein transfer by chaperones to proteasomes for both itself and other proteins.

### 2.3. Cytoskeleton Impairment: Intracellular Transport and Synaptic Transmission

The structural role of *HTT* is to combine microtubules with carrier proteins such as dynein and kinesin for transport pathways in the cell. The absence of this function in mHTT impairs intracellular trafficking, thereby having a strong effect on synaptic activity owing to a decrease in the transport of mitochondria, vesicles, and other structures along axonal microtubular pathways [35,36,37,38]. Thus, neurotransmission via receptors such as the GABAA (γ-aminobutyric acid receptor type A) and AMPA (α-amino-3-hydroxy-5-methyl-4-isoxazole propionic acid) receptors deteriorates because mHTT impairs the ability of the HAP1 protein to mediate the binding of the KIF5 protein to the receptors implementing microtubule transport [39,40]. In addition, the transport of neurotrophic factor BDNF and its receptor TRKB, which ensure the survival of neurons, are disturbed too [41].

Transport failures, coupled with a decrease in energy, also diminish the reuptake of glutamate (a ligand of NMDA receptors), as a result of which excitotoxicity develops. It consists of hyperactivation of NMDA receptors and an enhancement of the flow of calcium ions into the cell; this alteration triggers the respiratory chain of electron transport. Excessive exposure to calcium induces upregulation of reactive oxygen species [42].

### 2.4. Mitochondrial Dysfunction

Mitochondria are a link between several pathological processes: disturbances of the electron transport chain, of metabolic processes, and of calcium homeostasis [11,12].

For ATP synthesis, transmembrane potential of mitochondria should not deviate from the optimal value. A decrease in the activity of respiratory chain complexes, especially complex II (succinate dehydrogenase) and complex III (ubiquinol-cytochrome c oxidoreductase), as well as impaired protein transport between mitochondria and cytosol, owing to the interaction of mHTT with TIM23, cause an aberration of membrane potential of mitochondria [43]. Furthermore, mHTT provokes mutations in mitochondrial DNA, which lead to heteroplasmy [44].

Problems also appear in metabolic pathways because it is reported that mHTT inhibits the binding of TAF4 to the *PGC-1α* gene promoter, which is involved in glucose metabolism and fatty acid β-oxidation [45]. The GAPDH protein binds to N-terminal regions of mHTT, thereby providing its enhanced delivery to the nucleus, and subsequently loses its activity after entering the mHTT aggregates [46].

Normally, electron leakage from the electron transport chain occurs mainly due to the functioning of complexes I and III resulting in the formation of reactive oxygen species, such as the superoxide anion and peroxides, which the antioxidant system can handle. In addition, the activities of enzymes in the Krebs cycle, for example, aconitase, are worsened [47].

Mutant HTT not only enhances the formation of reactive oxygen species but also disrupts the expression of proteins of one of the pathways of the antioxidant system: NRF2/ARE [48]. As mHTT interacts with CREB-binding proteins and TAF4, mHTT decreases the amount of coactivator PGC-1α, which negatively affects the expression of respiratory chain enzymes’ nuclear subunits, of the antioxidant system, of TFAM (a mitochondrial–DNA transcription regulator), of ATP synthase, and of superoxide dismutase [49].

By binding the MFN2 protein, N-terminal fragments of mHTT disturb the expression of genes necessary for maintaining mitochondrial morphology and mitogenesis in *C. elegans* models of HD. In addition, the whole mHTT protein accumulates in mitochondria and stimulates division of mitochondria by recruiting Drp1. It was recently shown that inhibition of the calcineurin–Drp1 pathway or upregulation of Opa1 promotes the survival of striatal neurons [50].

Mitochondria-associated membranes are microdomains enriched with inositol-1,4,5-triphosphate (IP3) receptors (IP3Rs), ryanodine receptors (RYRs), and molecular chaperones (such as glucose-regulated protein p75) that form functional complexes with voltage-gated anion-selective channels to ensure the transfer of Ca^2+^ from the endoplasmic reticulum to mitochondria [51].

mHTT depolarizes the mitochondrial membrane in several ways: it affects the outer and inner pores of mitochondria and induces the formation of ion channels, which changes the flow of protons through the inner membrane of mitochondria. Membrane depolarization, in turn, disrupts calcium buffering in mitochondria and promotes the opening of permeability pores (MPTP) at lower Ca^2+^ concentrations, thus leading to a cytochrome c release and mitochondrial swelling. Another way to increase the calcium concentration in the cytosol is the effect of mHTT on IP3 receptors [12,52].

### 2.5. Cell Death 

Events taking place in mitochondria under the influence of N-terminal fragments of mHTT induce the release of cytochrome c, its binding to deoxy-ATP, APAF-1, and caspase 9, and formation of the apoptosome. This complex then activates caspases 3, 6, and 7, initiating caspase-dependent apoptosis [12]. 

Different caspases have dissimilar effects on the course of pathological processes: caspase 2 can cleave mHTT and generate toxic N-terminal fragments causing neurite degeneration; caspase 7 in a complex with mHTT activates other caspases, thereby stimulating apoptosis [53]; and caspase 8 cleaves off and activates the BCL2 (B-cell lymphoma 2) domain, whose activity induces caspase-independent apoptosis. mHTT also stimulates apoptosis in other ways: binding to the transcription factor p53 and increasing the expression of Bax and PUMA (p53-activated modulator of apoptosis) [54]. Additionally, the interaction of mHTT with HIP1 is weakened, which allows HIP1 to freely interact with proapoptotic proteins, for example, procaspase 8, which also provokes cell death.

Intensive production of reactive oxygen species caused by mitochondrial dysfunction results in DNA damage. Emergence of DNA double-strand breaks hyperactivates poly-ADP-ribose polymerase I (PARP1), which recruits DNA repair enzymes by synthesizing poly(ADP-ribose) chains [55,56]. Excessive accumulation of poly(ADP-ribose) triggers a process of cell death called “parthanatos”, during which proteins such as AIF and SMAC/DIABLO are released from mitochondria into the cytosol. AIF moves into the nucleus and causes DNA fragmentation and inhibition of poly(ADP-ribose) polymerase, thereby accelerating cell damage and destruction. SMAC/DIABLO binds to an X-linked apoptosis inhibitor protein and triggers apoptosis by inhibiting the antiapoptotic activity of the X-linked apoptosis inhibitor protein [56].

### 2.6. The Role of Astrocytes

Aggregates of mHTT are observed in astrocytes, although to a lesser extent than in spiny neurons [57]. Astrocytes are involved in the protection of neurons from excitotoxicity, in particular, by regulating the level of glutamate in the extracellular space through transporters such as EAAT2 and GLT1 [58,59]. In some cases, for example, with a certain number of glutamine repeats (polyQ), astrocytes exhibit signs characteristic of HD in neurons [60,61]. Inwardly rectifying K^+^ channels’ abundance is decreased in astrocytes, and therefore membrane potential diminishes as well [62]. Binding of mHTT to N-terminal myelin regulatory factor and inhibition of concurrent binding of CREB and TAF4 by mHTT reduce the expression of myelin core proteins and cause myelination deficiency and oligodendrocyte dysfunction [63].

### 2.7. Molecular Pathways of HD

Established pathways of HD can be found in databases of protein–protein interactions (PPIs) or molecular pathways and Gene Ontology. For instance, in the KEGG database (Kyoto Encyclopedia of Genes and Genomes, https://www.genome.jp/kegg/, accessed on 23 August 2023), there is information about the following pathways in this regard: Calcium signaling, Ubiquitin–proteasome system, Autophagy, Apoptosis, TNF signaling, Oxidative phosphorylation, Microtubule-based transport, and Transcription.

A comparison of these data with previously described pathways from recent literature reviews [12,52,64] indicates that there are several mechanisms in the pathogenesis of HD.

Mutations in the *HTT* gene lead to the loss of functions and to emergence of new types of activity of mHTT, the appearance of toxic N-terminal fragments during proteolytic activity, and mHTT aggregation. Collectively, these events disrupt the entire functional network of *HTT* and lead to failures in areas of natural functions of mHTT. In addition, due to the new functions of mHTT, N-terminal fragments, and the formation of physical obstacles in the form of protein aggregates, other cell processes fail too. In general, the following disturbed areas of cell physiology can be distinguished: cellular transport, systems of defense and homeostasis of the cell, and energy supply and metabolism (in particular, mitochondrial activities).

Cellular transport means the processes of physical motion of molecules or cellular structures, such as mitochondria, autophagosomes, or vesicles containing neurotransmitters. Structures of the transport system support both cell division (mitosis) and the mechanics of cell movement and migration, which involve the cytoskeleton and its associated proteins.

The second area of cell physiology that suffers because of mHTT is the systems of the cell response to stress, or defense systems, in particular, autophagy and various types of cell death (apoptosis and parthanatos) [65,66,67]. mHTT affects the defense systems not only at the cellular level but also in the whole body. Hyperactivation of activated microglia and cells of the peripheral immune system is induced by mHTT binding to transcription factors and leads to chronic inflammation and damage to neurons and peripheral tissues [68].

The consequences of impaired mitochondrial functions and their disturbed integrity are extensive. The aberrant activity of mHTT impedes mitochondrial biogenesis and results in a mitochondrial cytochrome c release, caspase activation, calcium dysregulation, and decreased energetic function [69]. Aside from a decrease in cell energy supply, (i) upregulation of reactive oxygen species, (ii) the induction of caspase-mediated apoptosis, and (iii) AIF-mediated parthanatos are among the key factors of neuronal death [66]. Based on the above facts, several basic pathways initiating mitochondrial dysfunction can be distinguished: the involvement of mHTT in the regulation of transcription, in transport of molecules into mitochondria, and in binding to enzymes and distortion of their functions. Pathological consequences of mHTT’s functions are summarized in Figure 1. 

Some of the concerning questions are mHTT’s protein interactors and related processes. Addressing these questions should clarify not only the mechanism of HD but also new therapeutic targets. Above, we reviewed some impairments during HD development; however, the extent of mHTT involvement in these processes remains uncertain.

Recently, Wanker et al. reviewed mHTT’s PPIs and related pathways, such as axonal transport, autophagy, mTOR signaling, palmitoylation, mitochondrial fission, mitophagy, and transcription regulation [70]. Greco with the research group have analyzed PPIs of the HTT protein containing normal or expanded CAG repeats in a mouse HD model [71]. They have demonstrated that some processes associated with mHTT depend on the polyQ length and revealed the most common pathways of mHTT: processes related to vesicular trafficking and synaptic signaling. Accordingly, it was hypothesized that mHTT sequesters the key proteins that are necessary for the regulation of synapse morphology and neurotransmission.

Another research article on mHTT PPIs—with some severe limitations, e.g., in terms of polyQ length, experimental methods, and HTT species—has revealed the following PPI clusters associated with mutant huntingtin: protein modification, RNA splicing, mitochondria, granule membrane, macroautophagy, cytoplasmic vesicles, ion channel transport, and translation [72]. Each cluster involves the top KEGG pathways related to huntingtin.

### 2.8. HD Treatment 

Today, most treatments reduce the motor and behavioral symptoms of HD; however, they do not address the underlying causes at the molecular level. In this section, we consider common therapeutic strategies targeting mHTT and pathological cell processes.

One of the strategies is to reduce the protein or mRNA level of mHTT. The latest therapeutic approaches are aimed at degradation or isolation of mRNA of mutant huntingtin, at suppression of its transcription, or at altering its processing [73,74]. Another way is to modify the mutant protein to decrease toxicity.

Antisense oligonucleotides (ASOs), by hybridizing with complementary mRNA, promote mRNA degradation and downregulation of both wild-type and mutant HTT. Testing on primates and rodents has yielded positive results [73], but such a therapy requires readministration for long-term suppression of mHTT synthesis. ASOs are also used for selective alteration of mRNA processing with the aim to prevent mHTT synthesis and cleavage and the formation of toxic N-terminal fragments [74]. To date, a number of ASO-based therapeutics have been developed and approved by the FDA: fomivirsen (Vitravene; for cytomegalovirus retinitis), mipomersen (Kynamro; for familial hypercholesterolemia), eteplirsen (Exondys 51; for Duchenne muscular dystrophy), and nusinersen (Spinraza; for spinal muscular atrophy).

Transcription repression is achieved also with the help of synthetic zinc finger protein (ZFP) repressors. These proteins consist of a zinc finger domain fused with some repressor, such as the KOX1 transcriptional repressor domain, and can selectively bind *HTT* containing expanded CAG repeats, thereby suppressing transcription. Testing of this therapy on the R6/2 model of HD has shown a reduced level of the mHTT protein and symptom stabilization [75]. It is necessary to conduct further trials to achieve greater selectivity and to check for side effects.

RNA interference machinery involving small interfering RNAs (siRNAs) [76], artificial microRNAs (miRNAs) [77], or short hairpin RNAs (shRNAs) [78] is also considered a potential HD therapy. Some siRNA- and miRNA-based therapeutics are in clinical trials, e.g., the AMT-130 trial (NCT04120493 and NCT05243017).

One more mRNA-targeting modality is small molecules mediating alternative splicing of mutant huntingtin mRNA: PTC518 (NCT 05358717, LMI070 (branaplam), NCT05111249).

The most direct way to decrease the mHTT level is to eliminate the mutant protein or to block its activity. Rapamycin, an inhibitor of the mTOR pathway, induces mHTT protein autophagy [79]. Cystamine and pridopidine are being tested to reduce protein aggregation. rAAV6-INT41 is an intrabody binding the polyP/proline-rich region of mHTT, thus reducing the aggregation in neurons of the R6/2 HD mouse model [80].

Another strategy of HD treatment is to target defective molecular pathways or functions. Generally, the following pathological processes are distinguished: impaired signal transduction, abnormal degradation of proteins, altered protein folding, transcriptional dysregulation, mitochondrial dysfunction [81], an axonal transport disturbance, and glial dysfunction [82].

Accordingly, based on currently known mechanisms of HD, numerous drugs are discussed in the literature. Firstly, a lot of studies are focused on synaptic dysfunction. To suppress excitotoxicity, drugs aimed at reducing glutamate receptors’ activity or availability of extracellular glutamate, e.g., Metamine [83] or BN82451 [84], have demonstrated efficacy in studies. Nonetheless, the safe concentration has not been determined yet. Another research field is aimed at regulating the dopamine level. FDA-approved drugs such as tetrabenazine [85] and valbenazine [86] inhibit vesicular monoamine transporter 2 (VMAT2), thereby reducing dopamine signaling. In addition, antibodies as a therapeutic modality are widely used: ANX005 (a monoclonal antibody) inactivates pathogenic complement cascade activation initiating neuroinflammation [87].

Mitochondrial dysfunction in HD has not been successfully corrected yet. Currently, resveratrol, which is an inhibitor of p53-mediated mitochondrial apoptotic oxidative stress [88], and PPARα agonist fenofibrate [89] are at the stage of clinical trials.

Most of described drugs in this field are intended to correct the molecular mechanisms whose malfunctions have been described. Anyway, a target can be identified among HTT’s or mHTT’s interactors directly: a loss of function or gain of function of mHTT may be an inducer of impairment. A low level of BDNF is associated with HTT dysfunction, leading to deficient synaptic transmission [90]. Histone hypoacetylation is one of the hallmarks of HD. mHTT can bind histone acetyltransferases, thus altering their functions. Sodium butyrate (a modulator of histone deacetylases) improves R6/2 mice’s motor symptoms and extends the lifespan [91].

Some researched drugs, such as neflamapimod [92] and minocycline [93], are utilized to ameliorate glial cells’ functions for preventing neuroinflammation.

Additionally, regardless of a target molecule, there is a type of therapy that solves the problem of degeneration through invasion. This method mainly concerns the maintenance of the population of medium spiny neurons. One of the approaches is to use stem cell-based therapy. Currently, researchers attempt to derive a pure population of a relevant type of cells not inducing an immune response: mesenchymal stem cells are at the clinical trial stage (NCT01834053) and induced pluripotent stem cell-derived neural stem cells have been tested on YAC128 HD mice [94].

Overall, the FDA has approved several drugs for HD. Among them, there are drugs designed to treat core symptoms, e.g., Austedo (deutetrabenazine), Xenazine (tetrabenazine), Valium, Risperdal (risperidone), and Ingrezza (valbenazine) (http://www.fda.gov/drugsatfda, accessed on 12 November 2023). Behavioral and cognitive abilities possess more complex mechanisms and regulations and have a variety of therapeutically targeted characteristics. Accordingly, to improve mental symptoms, drug repurposing is often used. FDA-approved drugs such as Clozaril (clozapine), Geodon (ziprasidone), Seroquel (quetiapine), Xanax (alprazolam), and Zyprexa (olanzapine) have originally been created for the treatment of schizophrenia or other mental disorders [95,96]. Some of these drugs have been approved for the treatment of HD, others are still at the clinical trial stage, but a search is still underway for a drug that could not only temporarily suppress symptoms or partially correct some molecular pathologies, such as aggregation, but has a long-term impact on multiple hallmarks of the disease with minimal adverse effects.

## 3. Drug Repurposing 

Drug repurposing is the process of reprofiling existing drugs (which have already been approved) to treat other diseases. Pharmacological properties of some existing drugs make them effective in the treatment of various diseases. This approach may lead to the discovery of new treatments for a range of diseases, aside from saving the time and resources expended on drug development.

Before starting a search for mHTT interactors, their molecular pathways, and drugs, let us review the existing approaches to the development of drugs on the basis of knowledge about molecular pathways. For example, many phenomena that accompany HD are also characteristic of other neurodegenerative diseases and other types of illnesses. For instance, oxidative stress is one of the causes of neuronal death in Alzheimer’s, Parkinson’s, and Huntington’s diseases, but reactive oxygen species also accompany oncological, infectious, and other pathological processes in the cell. Enhancing antioxidant protection against reactive oxygen species is therefore an evident therapy aimed at protecting cells from the oxidative stress caused by any pathological agent [97]. Of course, there are characteristic mechanisms and pathways that are specific only for a few diseases, and there is a reason to attempt to modify a pathology precisely through them. In this case, it is possible to address the problem at the root rather than eliminating consequences such as DNA damage and altered metabolic pathways, signal transduction, and other phenomena.

There are more narrowly targeted options for drug design where both consequences and root causes intersect in a disease under consideration. For example, PARP family repair enzymes (already mentioned above) are a potential target for HD therapy [2].

De novo drug development has many advantages, is aimed at a disease specifically, and typically relies on a detailed pathogenesis analysis that accompanies drug development and validation. On the other hand, de novo development takes a lot of time and other resources: from the stages of target selection, design and optimization of a drug’s structure, and investigating the methods of drug delivery to preclinical and clinical trials. The whole process can take up to 10–20 years. 

Repurposing or reuse of drugs that have proven effectiveness against other diseases helps to save time spent on developing the drug itself and on adjusting and optimizing its design. 

Both de novo drug development and retargeting of existing drugs share the same problem: target selection. Several approaches have been proposed for target identification. The choice of strategy depends on available initial data. Genomic and proteomic data such as gene expression patterns or protein abnormalities associated with a disease can reveal a potential drug target. Analysis of the interaction of proteins with small-molecule compounds is an affinity-based approach [98]. If we know the molecular mechanism of the corrected process, then a specific molecule could be the target, and the treatment outcome can manifest itself in phenotypic changes (at the cellular level: e.g., a decrease in protein aggregation). The phenotype-based approach involves the testing of new potential therapeutic substances in parallel with drugs having a known effect [99,100,101].

In the case that the genes or biomolecules involved in a pathological process are known, it is possible to take advantage of an analysis of the interactions between involved molecules and drugs in a database such as DrugBank [102]. Furthermore, it is feasible to start searching for drugs through molecular pathways by choosing the most significant ones for the disease of interest by means of such databases as Reactome and ShinyGO 0.77 [103]. Another option is to search through a database with ready-made collections of descriptions of mechanisms underlying cellular processes where the descriptions connect them to specific diseases. Text mining and neural network technologies are indispensable for finding molecular pathways and for collating them with drugs and known disease characteristics (genes, molecular processes, and molecular phenotypic features) [104]. This approach derives from the fact that we do not know much about molecular processes of the disease, which is usual for rare diseases. Then, we turn to a search by clinical similarity of diseases and collect data to reconstruct mechanisms of the disease and to predict the effects of drugs. 

There are a few areas of drug repurposing reviewed in the literature: drug oriented, disease oriented, and target oriented [105,106]. The first approach involves collating a known drug with new target molecules. It includes studies on topics such as off-label use of drugs, phenotypic screening, targeting 3D structures of a drug, chemical structures of drugs and ligands, and adverse effects of drugs. Disease-oriented drug reprofiling is employed for a number of diseases on the basis of a similarity of their characteristics (phenotypes and gene profiles), disease pathways, disease ‘omics data, genetic data on a disease, and a protein interaction network. Target-oriented drug repurposing includes identifying new targets in the pathogenesis of the disease and testing the impact of a known drug associated with a found target. It is possible to use disease ‘omics data and information related to treatment strategies, genetics, proteomics, and metabolomics. These strategies share methods of repurposing such as a blinded search or screening method, target-based methods, knowledge-based techniques, signature-based methods, pathway- or network-based procedures, and targeted mechanism-based methods. 

A recent publication highlights strategies used in drug development and proposed for the repurposing of drugs for neurodegenerative diseases [105]. These approaches are as follows: 1.Activity-based or experimental drug repurposing [107].

Experimental testing of the effects of a drug is considered if there are few data on the mechanism of action. In addition, a direct response to a therapeutic is usually a consequence of a similarity between the reference disease and target disease, and the effect is analyzed further [108]. The targeted approach is another strategy of activity-based drug repurposing that involves knowledge of molecular targets [109].

2.Computational drug repurposing.

Computational tools can be classified into the following categories:a database with drug data: phases of clinical trials, a mechanism of action, involvement in diseases, and physicochemical properties;a database with disease data: genes, molecular pathways, and mechanisms;tools for the analysis of molecular interactions (construction of gene, metabolic, and protein networks), Gene Ontology, and analysis of transcriptome data;tools of molecular dynamics and docking: construction and analysis of structures of target molecules and of the drug as well as interactions;text-mining tools, machine learning (ML), and neural networks.

Structure-based or target-based methods such as molecular docking and similarity-based approaches are aimed at finding drugs with strong affinity for selected targets. This approach is good if we have already chosen certain target molecules with known structures and physicochemical properties, and we are looking for suitable compounds for them, based on their structure and thermodynamic characteristics.

Ligand-based methods are useful when there is no information about the 3D structures of potential pathogenesis-related molecules; in this case, we choose a drug on the basis of information about ligands interacting with the target, thereby obtaining information and physicochemical properties of target molecules.

Transcriptome-based methods make use of extensive new-generation sequencing data. Changes in gene expression profiles under the influence of various substances are used to select potential targets. For example, there are databases such as CMap (Connectivity Map) [110] and LINCS (the Library of Integrated Network-Based Cellular Signatures) [111] containing data about effects on a gene expression profile in various tissues and drug-treated cell lines. 

3.Genome-wide association study (GWAS)-based methods.They identify a gene variant–disease relation, which subsequently helps with the selection of treatment targets.4.Network-Based Methods.These procedures include system biology approaches for integrating and analyzing data on relations between various objects: an interaction of cellular structures with each other and with drugs under various conditions, including during the progression of a disease.5.ML-based approaches and literature-based discovery methods.ML offers methods for regression analysis, clustering and classification, dimensionality reduction, neural networks, and other tools helping to analyze biological data and to infer new trends [112]. ML is now actively utilized at various stages of drug design: from investigating disease mechanisms, target identification, target validation, and compound screening to finding new markers of drug efficacy.

For identifying genes of potential targets associated with a disease or phenotype, Costa et al. (2010) invented a decision tree-based meta-classifier trained on network topologies of PPIs and of metabolic and transcriptional interactions [113]. Another area of ML applications is the optimization of drug–target compatibility via screening of similar ligand structures on the basis of a reference. This is achieved through multitask deep neural networks.

The literature-based approach includes automated processing of the literature, e.g., natural language processing methods for text mining of specific information. For instance, the BeFree tool applies natural language processing techniques to scanning drug–disease, gene–disease, and target–drug connections in Medline abstracts [114]. Yang et al. (2017) presented a pattern-based learning method for extracting from the biomedical literature potential drugs for repositioning [115].

We combined the aforementioned approaches of drug repurposing and devised a two-step strategy (Figure 2) basically aimed at the database search approach to drug repurposing [116,117]. First, we should receive information about molecular participants and disease processes. This can be achieved via experimental methods and in silico methods such as text mining and analysis of drug databases, PPI databases, and disease databases. The mechanism of a disease may not have already been studied. For this purpose, we can use additional tools to analyze the relations of disease-associated molecules.

Next, it is necessary to connect disease genes, proteins, and pathways to known drugs. The simplest way is to search for a drug that targets certain pathogenesis-inducing molecules. Nevertheless, to understand the effect of the drug, molecular pathways of molecular targets should be analyzed too. In addition, the second method of drug repositioning is the search for pathway-associated drugs. There is a list of databases helpful for the search for a drug (Figure 3). Ultimately, an investigator needs to simulate the effect of the drug. To this end, it is worth knowing the pathogenesis of the disease, not just individual molecular pathways (Figure 4).

## 4. Drug Repurposing for HD Treatment

Numerous articles are devoted to the development of approaches to drug repurposing for rare orphan diseases, which include HD. In relation to HD, ideas for reuse of approved drugs appeared in 1997 [118], when clozapine was tested in a double-blind randomized trial on patients with HD. In subsequent years, trials of olanzapine [119], memantine [120], tetrabenazine [121], risperidone [122], tetrabenazine [123,124,125], cysteamine bitartrate [126,127,128,129,130,131], lithium citrate [132,133,134,135], laquinimod [136,137], rapamycin [138,139,140,141], minoxidil [142], felodipine [143], nilotinib [144,145], dextromethorphan [146], and INO-1001 [147] have been conducted. Ref. [148] (2021) provides examples of approved and unapproved drugs repurposed for the treatment of chorea in HD: tetrabenazine (Xenazine^®^) [149] and Austedo (deutetrabenazine), which is a compound with VMAT2 inhibitor activity [150]. Not yet approved substances include cysteamine bitartrate and lithium citrate tetrahydrate. In mouse models of HD, cysteamine bitartrate has been found to extend the lifespan and reduce motor impairment. Nevertheless, when tested on patients with HD, it did not give positive results [127]. It is thought that the drug slows the progression of the disease in patients with more severe types of dysmotility. Lithium citrate tetrahydrate in preclinical trials has helped to reduce the formation of intracellular protein aggregates and improved motor activity and coordination of movements [132,151]. In short-term clinical trials, the drug has been shown to improve motor activity and reduce symptoms of chorea [152,153]. 

Another drug, rapamycin (sirolimus, Rapamune), activates the autophagy system by inhibiting mTOR [139,141]. Despite the positive effect, rapamycin has other activities; therefore, research has been carried out to find highly specialized drugs that stimulate autophagy: clonidine, verapamil, loperamide, nimodipine, and minoxidil have been identified in this way; all of them inhibit the aggregation of mHTT in neuroblastoma cells in vitro [143].

In addition to disturbances in the systems of proteolysis of abnormal protein forms, in HD, there is an abnormal activation (induced by the CNS) of the immune system and a weakening of the activity of immune cells in the periphery [154]. Thus, it has been decided to use the drug laquinimod developed for multiple sclerosis. In primary neurons derived from a mouse model of HD (YAC128), in vitro, this drug reduced the level of apoptosis, and, when tested in vivo on mice, it improved motor and mental activity and diminished the level of IL-6 in serum [137]. In second-phase clinical trials, there is a significant reduction in caudate atrophy in patients with early HD [137,155].

Two other repurposed drugs are in clinical trials. The first one, nilotinib (approved for the treatment of chronic myeloid leukemia), has earlier been successfully used in the treatment of Parkinson’s disease and is believed to reduce the accumulation of mHTT [144,145]. The second drug, Nuedexta (previously utilized for the treatment of the pseudobulbar affect), has been shown to significantly alleviate agitation in patients with Alzheimer’s disease [146].

Another work (2018) generalizes repurposed drugs for a number of neurodegenerative diseases, including HD [156]. Tetrabenazine, first used as a drug with antipsychotic activity, acts as a reversible inhibitor of monoamine reuptake by presynaptic neurons. As it turned out, the drug proved to be successful in the treatment of HD, and, by analogy, other drugs with dopamine antagonistic activity have been found: tiapride [157] (a D2 receptor antagonist), clozapine [158] (a dopamine D1 and D4 receptor antagonist, although clinical trials have yielded conflicting results [118]), olanzapine [119] (which has strong affinity for serotoninergic receptor and for D2 receptor [159]), risperidone [122] (a D2 receptor antagonist and a serotonin agonist), quetiapine [153] (which has affinity for serotonin and dopamine receptors), and memantine [83] (which can prevent calcium influx into neuronal cells thereby preventing cerebral cell death and can decrease the vulnerability of neurons to glutamate-mediated excitotoxicity [160]).

FDA-approved drugs for HD in use include tetrabenazine and deutetrabenazine for symptoms of chorea, risperidone and olanzapine as antipsychotics, citalopram, fluoxetine, sertraline, lamotrigine, and carbamazepine (the last two as mood stabilizers) [161].

PARP inhibitors are novel potential therapeutic agents for HD. It is known that PARP1 protects neurons from cell death under mild oxidative stress by promoting DNA repair [3]. This repair enzyme is reported to be hyperactivated in neurons of patients with brain pathologies, and the involvement of repair pathways in the development of HD has been demonstrated [162,163,164,165]. This enzyme participates in the induction of cell death, called parthanatos, which is mediated by the AIF protein [166]. The enzymes PARP1 and PARP2—being participants in DNA strand break repair, chromatin regulation and transcription, cell cycle, metabolic regulation, inflammation, and activating a cell death pathway called parthanatos—are actively employed as a target for prevention of neurodegeneration [167,168,169]. DNA repair proteins interact with poly(ADP-ribose), which is important for loading DNA repair enzymes onto sites of DNA damage [170,171]. Thus, normally, PARP rescues the cell in the event of DNA strand breaks by recruiting DNA repair proteins (for example, DNA polymerases and XRCC1) to the damage site. Nevertheless, prolonged damage to DNA integrity, coupled with events accompanying pathological processes, reduces the ability of PARP to use ATP and NAD^+^, owing to depletion of their pools and provokes cell necrosis and PARP hyperactivation inducing parthanatos [3,165,166,167,168,172,173]. Drugs based on PARP1 inhibitors have been created to treat cancer, but, as it turned out, similar molecular pathways made it possible to expand the scope of their clinical application. 

The idea of using PARP inhibitors for the treatment of neurodegenerative diseases has appeared quite a long time ago [168]. A group of researchers in Ref. [174] revealed a connection of *HTT* with the ATM protein of the DNA repair pathway, where ATM plays the scaffold role, and this connection prevents the repair process. Furthermore, they published reviews in 2019 and 2021 on the participation and mechanisms of DNA repair and PARP enzymes in HD [162,175].

The INO-1001 compound has been tested on R6/2 mouse models of HD featuring expression of an N-terminal part of mHTT and of the full-length protein, which have made it possible to observe its interaction with poly(ADP-ribose)-tagged proteins [147]. Although the mechanism behind the involvement of PARP enzymes in the HD pathogenesis is still not entirely clear, it is known that there is an influence of mHTT on PARP activity. Owing to a positive effect of inhibitors INO-1001 [147] and Olaparib [176] on mice featuring the expression of *HTT* exon 1 with an expanded polyglutamine tract, efforts can be directed to further investigation into the impact of PARP inhibitors on the progression of HD in more relevant models such as cell lines derived from HD patients.

## 5. Drug Repurposing Targeting mHTT Interactome Pathways

The most important and most difficult task in the search for a target is reconstruction of the disease mechanism in order to optimize the action of the drug. Even when the search for drugs is based on indirect signs (disease hallmarks), it is necessary to reproduce the mechanism along the way. After examination of “drug repurposing” in general, the next step is discovery of protein targets and HD molecular pathways for the reuse of approved drugs from databases.

We found a set of genes in the mHTT interactome and identified their molecular pathways, which is necessary to identify associated drugs (Appendix A, Appendix A). For this purpose, we searched different sources related to HD studies and found 490 genes in the HTT-OMNI database [177], the PANTHER database [178], and mHTT interaction studies in human cells: 276 genes were discovered in the Podvin (2022) study [72], and six other genes—*CCT2*, *PRMT6*, *KMO*, *JPH1*, *STIM1*, and *TFEB*—came from later papers dealing with experimental and bioinformatic research of HD in years 2022–2023 [179,180,181,182,183,184,185,186,187,188,189,190,191,192,193,194,195,196,197,198,199,200,201,202,203,204,205,206,207,208,209,210,211,212,213,214,215,216,217,218,219,220,221,222,223,224,225]. 

We analyzed the molecular pathways corresponding to the found genes to understand which processes mHTT targets (Appendix A).

In the Reactome database [226], we selected all pathways with a *p*-value > 10^−5^ (Appendix A) and ranked them in terms of significance. The most significant (at threshold *p*-value < 10^−10^) are the following pathways: the cellular response to stress, programmed cell death, signal transduction, vesicular transport, regulation of the cell cycle, and transcription. 

As follows from the aforementioned causes of the development of neurodegeneration, the regulation of transcription requires special attention. Therefore, let us consider in more detail what can be gleaned from the data on gene expression.

We found that PARP1 is involved in the regulation of transcription factors TP53, RUNX1, and RUNX3, where it intersects with mHTT. TP53 is a transcription factor that is a tumor suppressor; mutations in its gene cause malignant transformation of cells. On the other hand, the facts of its participation in the pathogenesis of neurodegenerative processes are also known, in particular, its mutation upregulates reactive oxygen species [227]. Transcription factor RUNX1 takes part in hematopoiesis, angiogenesis, and neuronal development. In addition, it is expressed in adults in hippocampal cells and participates in neurogenesis [228]. Transcription factor RUNX3 is responsible for the development of proprioceptive sensory neurons that support feedback from the CNS and returning signals about the state of the motor system [229]. For the above-mentioned processes associated with these genes, there are the following drugs that have passed clinical trials and possess known mechanisms of action. Carfilzomib (DB08889) is known for the RUNX1 pathway and is a proteasome inhibitor that is used to suppress the progression of multiple myeloma [230]. The molecular pathways of the *TP53* gene match drugs such as everolimus (DB01590), temsirolimus (DB06287), and sirolimus (DB00877). These compounds are mTOR inhibitors and cause decreases in cell proliferation, angiogenesis, and glucose uptake. They are used as immunosuppressors and tumor suppressors [231]. For instance, temsirolimus is employed to treat renal cell carcinoma because this substance causes G1 phase arrest in tumor cells.

Next, we used the KEGG [232] database to generate a process map of HD (map05016) and compared the proteins of this map with the mHTT interactome by means of the ShinyGO software for integrating molecular pathway data [103]. We aligned the processes—previously found by us in the Reactome database for the mHTT interactome—with the processes for HD on the map (Figure 5). It turned out that the list of processes for mHTT and its interactors is wider than that for HD.

We can search for a drug target by looking for processes–inducers or processes–consequences. The former means the direct contacts of mHTT that induce pathological changes in molecular pathways. By contrast, processes–consequences are associated with disease hallmarks such as DNA damage and repair, glutamate signaling, cell death, the antioxidant system, and others. Generally, all of the pathological processes result from a cascade of events caused directly by mHTT. Our findings represent an attempt to select a drug among root causes, i.e., initiating events. 

We used the list of mHTT interactome genes to search for drugs in the DrugBank database [102]. Assuming that a specific gene can be targeted by a pharmacological agent while knowing effects of this gene product on a molecular pathway, we can potentially utilize it to treat HD. This approach is facilitated by many of the facts discussed above, which point to a connection of neurodegenerative diseases with many others, in particular, with cancers, for which a wide variety of drugs have already been tested and approved (Appendix A).

Overall, we identified approved drugs (Appendix A) for the mHTT interactome (Figure 6). We also used the Gene2drug database [233] to find pathways and drugs on the basis of these pathways by means of the list of mHTT interactor genes. Nonetheless, the found drugs were too redundant, and we decided to try the algorithm presented earlier.

## 6. Conclusions

HD as a drug discovery field offers a variety of approaches, among which we focused on drug repurposing in this paper. This area has stood the test of time and has advantages over high-cost approaches; moreover, it allows for the use of accumulated knowledge about disease molecular mechanisms, drugs, and possible targets in conjunction with novel data-processing technologies, such as ML, neural networks, multifunctional software tools for integrating and analyzing molecular processes, construction of networks, and molecular dynamics simulations. This tool set facilitates reconstruction of cellular events and speeds up a drug search.

We mentioned one of the ways to use knowledge about a disease for drug discovery: on the basis of known molecular pathways and associated genes, we can look for the most appropriate targets among them while being guided by known pathways of the disease and analyzing the disease course under the action of each drug. Nonetheless, drug effect prediction must be reinforced by simulation of physicochemical characteristics in silico and in vivo, i.e., during clinical trials. HD partially simplifies such drug design: we employ the mHTT interactome for finding drug targets, simultaneously implicating target pathways, to clarify how a drug fits into the molecular network of the disease.

In this way, we compiled an mHTT interactome by searching databases and studies dealing with finding HD participants. Next, we found the interactome’s molecular pathways and determined genes that are targets of existing drugs. We revealed key pathways containing the proteins that have drugs targeting them: cellular responses to heat and chemical stress; transcriptional regulation by TP53, RUNX1, and RUNX3; apoptosis via activation of BH3-only proteins; and autophagy (mito- and aggrephagy).

DNA repair pathways involving PARP enzymes play a major part in HD pathogenesis. Comparing molecular processes between the mHTT and PARP interactomes uncovers novel overlapping pathways that can be considered supplementary therapeutic targets.

At this point, the found drugs are only raw materials for a more detailed study by computer simulation of their actions (taking into account the physicochemical factors that accompany HD progression), followed by experimental testing and comprehensive data analysis.

## Figures and Tables

**Figure 1 ijms-24-16798-f001:**
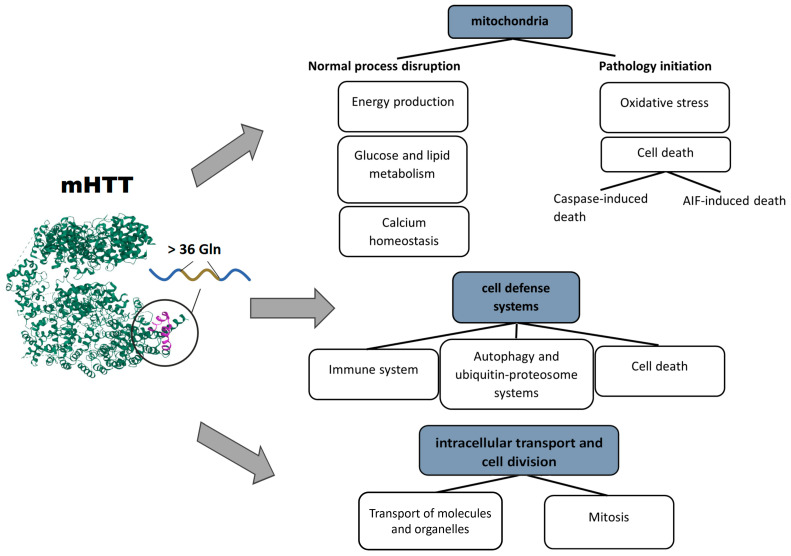
Pathological processes triggered by mHTT: mHTT causes a range of abnormalities in mitochondria, cell defense systems, intracellular transport, and cell division.

**Figure 2 ijms-24-16798-f002:**
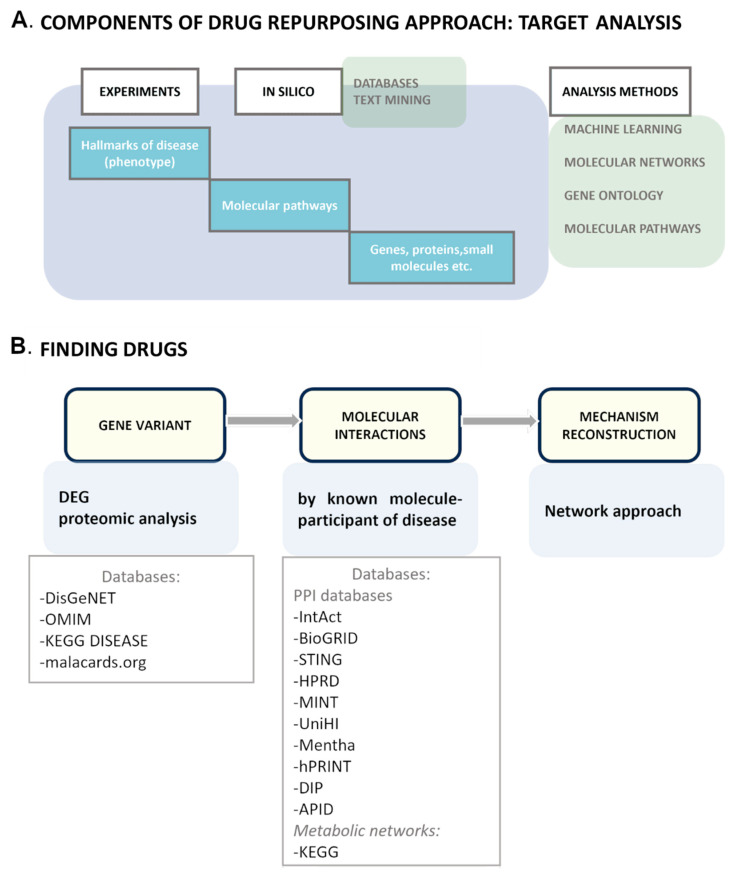
The drug repurposing pipeline. (**A**) The initial step of drug discovery is disease description: from phenotype characterization to identification of molecular mechanisms. There are two methods of target identification: experiments if molecular processes are not clear or a database search in combination with tools for analyses of connections between molecular or cellular processes. (**B**) To select a target, we should reconstruct a disease development map by analyzing known disease data and filling the gaps and then by determining processes and molecules connections. Even if the data are known, as a rule, obtaining a full consistent picture of a disease, especially in relation to existing drugs, is great luck. DEG: differential expression genes.

**Figure 3 ijms-24-16798-f003:**
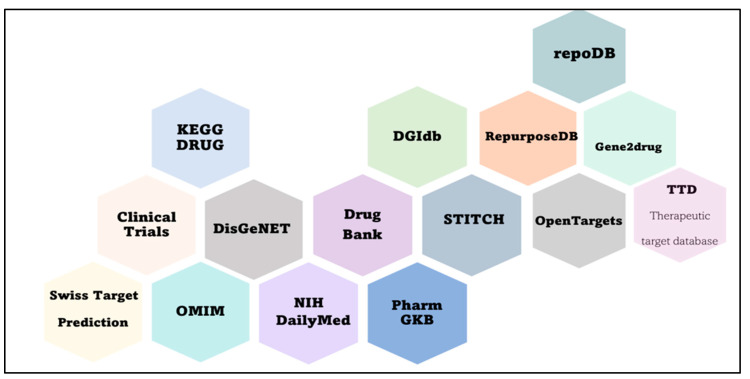
Databases for repurposed-drug discovery.

**Figure 4 ijms-24-16798-f004:**
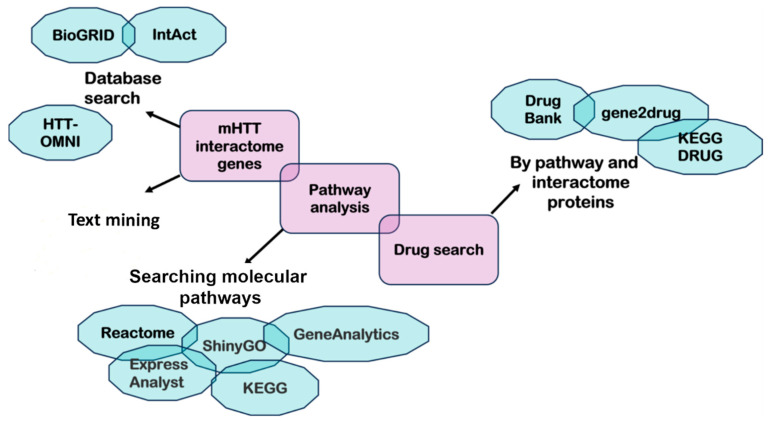
The drug repositioning algorithm based on a molecular pathway. The information about mHTT interactors can be obtained via in silico methods such as text mining and a database search. Identifying molecular pathways is necessary to link disease genes, proteins, and pathways to known drugs. The search for pathway-associated drugs can be performed in such databases as DrugBank, gene2drug, and KEGG DRUG.

**Figure 5 ijms-24-16798-f005:**
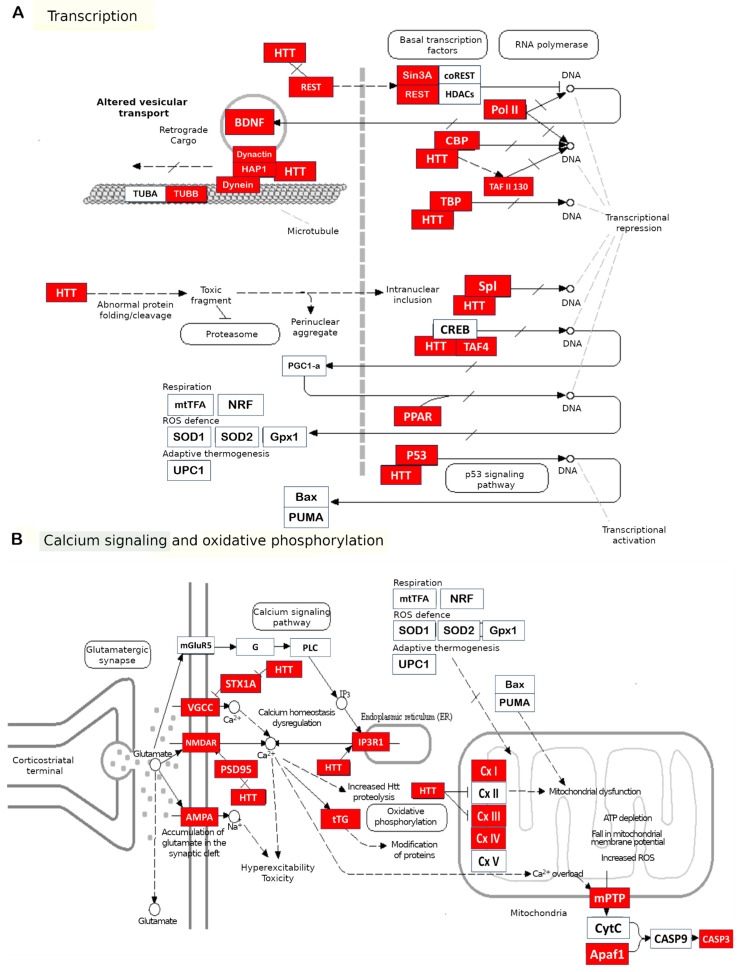
mHTT interactome genes (red) on the HD map (KEGG) and mHTT interactome pathways. The bottom box (**G**) contains the molecular pathways (in general form) that we found for the mHTT interactome. We aligned them to the map of HD from the KEGG database, labeled with the corresponding colors (**A**–**F**). Adapted from the KEGG database [232] with permission.

**Figure 6 ijms-24-16798-f006:**
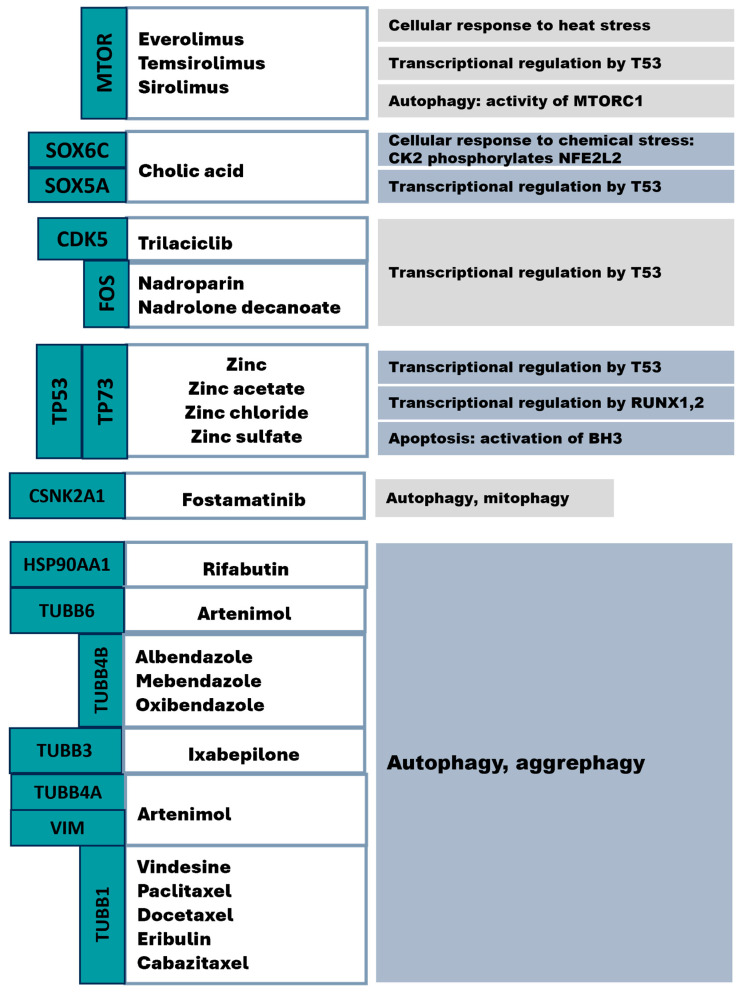
Approved drugs targeting members of mHTT interactome pathways (according to the Reactome database).

## Data Availability

Data are contained within the article and Appendix A. Raw data are available upon request.

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
