# Peer review of "Mutant-Huntingtin Molecular Pathways Elucidate New Targets for Drug Repurposing"

_ijms, 2023, doi:10.3390/ijms242316798_

Round 1

Reviewer 1 Report

Comments and Suggestions for Authors

In this work, the authors analyze genes and molecular pathways involved in Huntington's disease pathogenesis in order to describe the action of repurposed drugs and try to find new targets for therapy

Some criticism can be addressed to the paper:

a) The section “Drug repurposing” is hard to read and should be deeply revised, keeping in mind the target of the review. 

b)The authors report one of the ways to use knowledge about disease for drug discovery- based on known molecular pathways and associated genes, and  revealed key pathways, including protein  with targeted drugs, but readers also expect a comparison with other research in the literature. 

c) Some figures are of low quality (see figures 2, 5, 6)

Comments on the Quality of English Language

Moderate editing of English language required

Author Response

Dear Editors and Reviewers,

We really appreciate all the Reviewers comments! We revised the manuscript according to the comments and suggestions. Please see below our point-by-point answers. We also asked for English editing and now we hope that the manuscript was improved.

Reviewer 1

Comments and Suggestions for Authors

In this work, the authors analyze genes and molecular pathways involved in Huntington's disease pathogenesis in order to describe the action of repurposed drugs and try to find new targets for therapy

Some criticism can be addressed to the paper:

  1. a) The section “Drug repurposing” is hard to read and should be deeply revised, keeping in mind the target of the review. 

Thank you for the critical comments. We have revised the manuscript according to the reviewers’ recommendations. We especially paid attention to the section “Drug repurposing”. This section has been revised and reworded.

  1. b) The authors report one of the ways to use knowledge about disease for drug discovery- based on known molecular pathways and associated genes, and revealed key pathways, including protein with targeted drugs, but readers also expect a comparison with other research in the literature. 

We have added discussion of the literature data to the section on mHTT molecular pathways.

  1. c) Some figures are of low quality (see figures 2, 5, 6)

Sorry. The figures have been replaced with the sharp ones.

Moderate editing of English language required

The English language was corrected and certified by shevchuk-editing.com.

Sincerely,

Nadezhda Dyrkheeva and coauthors

Reviewer 2 Report

Comments and Suggestions for Authors

In my opinion, the aim of this review is attractive and points the light on an interesting topic. However, some comments needed to be addressed to improve the quality of the work prior to its publication. 

I found the “Drug repurposing chapter” difficult to read and incomplete. The bullet list's rationale is unclear and confounding. I suggest clarifying the schematization choices. Moreover, the list of the target identification approaches is not complete, there are more than the two listed. My suggestion is to make this chapter more complete and, if some aspects of drug repurposing are not covered in depth to state it clearly. 

“Drug repurposing in Huntington’s disease treatment” chapter is also incomplete and partial. Why are the genetic approaches not included? (e.g. antisense oligonucleotide, RNA interference, etc). If there is a specific reason for this choice, I suggest stating it in the review. 

Acronyms used in the text are not always made explicit in full. There is no concordance in the use of names (e.g., PGC-1α is also called PGC-1a; cytochrome c is also called cytochrome C). The use of capital letters for genes and proteins is not always correct, nor is the use of capital letters when referring to non-brand drugs (e.g. Memantine, Clozapine).

The images have low quality, and often appear grainy, cropped, unaligned, and difficult to read and understand (Figures 2, 3, 5, 6). Figure legends are not always clear, and more guidance should be given. 

The use of references is sometimes inaccurate I suggest putting more attention on the literature search. Reference number 46 is a retracted paper. Another reference must be found to justify the sentence. Reference 66-67/ 136/118 are double. Reference 80 is from bioRxiv. 

Comments on the Quality of English Language

I suggest reviewing the paper with a native English speaker.

Author Response

Dear Editors and Reviewers,

We really appreciate all the Reviewers comments! We revised the manuscript according to the comments and suggestions. Please see below our point-by-point answers. We also asked for English editing and now we hope that the manuscript was improved.

Reviewer 2

Comments and Suggestions for Authors

In my opinion, the aim of this review is attractive and points the light on an interesting topic. However, some comments needed to be addressed to improve the quality of the work prior to its publication. 

I found the “Drug repurposing chapter” difficult to read and incomplete. The bullet list's rationale is unclear and confounding. I suggest clarifying the schematization choices. Moreover, the list of the target identification approaches is not complete, there are more than the two listed. My suggestion is to make this chapter more complete and, if some aspects of drug repurposing are not covered in depth to state it clearly. 

The section “Drug repurposing” has been revised and reworded. We tried to correct the bullet lists and to make other changes according to this comment.

“Drug repurposing in Huntington’s disease treatment” chapter is also incomplete and partial. Why are the genetic approaches not included? (e.g. antisense oligonucleotide, RNA interference, etc). If there is a specific reason for this choice, I suggest stating it in the review. 

Thank you for your comment. Indeed, genetic approaches are important part of HD treatment. We have added the description of existing and developing methods of HD treatment as a separate section (2.8). However, the section “Drug repurposing in HD treatment” describes examples of repositioning of the recently approved drugs for HD treatment.

Acronyms used in the text are not always made explicit in full. There is no concordance in the use of names (e.g., PGC-1α is also called PGC-1a; cytochrome c is also called cytochrome C). The use of capital letters for genes and proteins is not always correct, nor is the use of capital letters when referring to non-brand drugs (e.g. Memantine, Clozapine).

Thank you. It was corrected.

The images have low quality, and often appear grainy, cropped, unaligned, and difficult to read and understand (Figures 2, 3, 5, 6). Figure legends are not always clear, and more guidance should be given. 

Sorry. The figures have been replaced with the sharp ones. The figure legends were revised.

The use of references is sometimes inaccurate I suggest putting more attention on the literature search. Reference number 46 is a retracted paper. Another reference must be found to justify the sentence. Reference 66-67/ 136/118 are double. Reference 80 is from bioRxiv. 

The reference list has been corrected. Duplicates and inappropriate citations have been removed.

Comments on the Quality of English Language

I suggest reviewing the paper with a native English speaker.

The English language was corrected and certified by shevchuk-editing.com.

Sincerely,

Nadezhda Dyrkheeva and coauthors

Round 2

Reviewer 2 Report

Comments and Suggestions for Authors

After the authors' revision, in my opinion, the manuscript is ready for publication.